# Efficacy of Lajjabati (*Mimosa pudica*) and Daruchini (*Cinnamomum verum*) extracts on wound healing in rabbits

Rukhsana Amin Runa 1*, Md. Abdur Rahim Prodhan[1], Suravi Akter[1], Afrina Mustari[2]

1 Department of Surgery and Obstetrics, Bangladesh Agricultural University, Mymensingh, Bangladesh,
2 Department of Physiology, Bangladesh Agricultural University, Mymensingh, Bangladesh

* ramin.so@bau.edu.bd

## Abstract

Plant and herbal preparations are traditionally used in wound management to promote healing. The study aimed to explore the therapeutic efficacy of Lajjabati (*Mimosa pudica*) leaves and Daruchini (*Cinnamomum verum*) paste on wound healing and the histo-architectural changes in the wounded skin of rabbits. Sixteen rabbits, weighing between 1.5 and 2 kg, were divided into four groups: A, B, C, and D, each containing 4 rabbits. Surgical incisions were made on the skin of the rabbits, creating a total of 32 wounds, two for each rabbit. Lajjabati (*Mimosa pudica*) leaves, Daruchini (*Cinnamomum verum*) paste, and a combination were applied to the skin wounds in Groups A, B, and C, respectively. Group D was considered the control. The morphological characteristics of wound healing, such as the swelling of sutured areas, elevation of the suture line, width of the sutured area, and contraction length, were recorded from Day 0 to Day 21. Bacteriological and histopathological samples were collected on Day 3, 7, and 14 for analysis. The swelling of the sutured area and elevation of the sutured line were $11.50 \pm 0.13$ and $2.54 \pm 0.10$ mm in Group A, which were significantly lower compared to other groups. The histopathological study revealed the presence of marked inflammation, hyperplasia, and enlargement of glands in Groups B and C, whereas in Group A, all tissues appeared to be normal, and hair follicles started to grow on Day 7. In microbiological study, the lowest bacterial colonies were observed in Group A. It is concluded that Lajjabati (*Mimosa pudica*) paste is more effective in the wound healing process. Daruchini (*Cinnamomum verum*) can also be used, but it causes more tissue reactions, indicating delayed healing of cutaneous wounds.

## Introduction

A wound is any disruption or damage to living tissue, such as skin, mucous membranes, or organs [1]. Wounds can occur suddenly as direct trauma by mechanical,

**Data availability statement:** All relevant data are within the manuscript and its Supporting Information files.

**Funding:** We acknowledge the National Science and Technology (NST), Ministry of Science and Technology (MoST), Government of the People's Republic of Bangladesh, for financial support. However, the funders had no role in study design, data collection and analysis, decision to publish, or preparation of the manuscript.

**Competing interests:** The authors have declared that no competing interests exist.

thermal, or chemical means or appear gradually over time. It can also lead to a basic break in the skin's epithelial integrity, and this condition can be worsened by damage to blood vessels, muscles, tendons, bones, and nerves [2]. So, wound management aims to regain functionality and physical well-being while reducing the risk of deformities and infections [3]. Wound healing is not a simple phenomenon; rather, it involves a complex interplay between numerous cell types, cytokines, mediators, and the vascular system [4]. The typical stages in the wound healing process involve inflammation, angiogenesis, granulation tissue formation, repair of connective and epithelial tissues, and the eventual remodeling of the tissue surrounding the injury site [5]. Antibiotics, analgesics, and herbal medicines speed up the healing of wounds [6,7]. Topical antibacterial therapy can modify the wound's environment by providing antimicrobial action, reducing fibroplasia, or promoting epithelialization [8]. Nutritional supplements as well as antibiotics significantly improve wound healing [9,10]. Several biomaterials, proteins, antibiotics, vitamins, and minerals appear to encourage angiogenesis, fibroblastosis,and wound epithelialization and expedite wound healing [11]. Plant-based antibacterial agents have been of great interest recently [12,13].

Different plants and herbal treatments are being utilized as a tradition to hasten the healing process of wounds. Practitioners of traditional herbal medicine have provided explanations for the therapeutic benefits of certain native plants for a range of illnesses. Due to their abundance as a rich source of ethnomedicine, medicinal herbs locally known as Lajjabati/Touch me not (*Mimosa pudica*) and Daruchini (*Cinnamomum verum*) are utilized traditionally throughout Bangladesh. *Mimosa pudica* has different pharmacological properties, including anti-inflammatory, antimicrobial, wound healing, analgesic, antioxidant, anticancer, and anti-proliferative activities [14]. Tannins, flavonoids, terpenoids, saponins, sterols, alkaloids, and phenols are some of the plant constituents revealed by the phytochemical screening of *M. pudica* [15]. The extract of Lajjabati (*Mimosa pudica)* reveals early epithelialization and wound contraction [16,17]. Eugenol, cinnamaldehyde, cinnamyl acetate, copane, and camphor are the major components of the *Cinnamomum verum* plant [18]. Eugenol is the active principal ingredient linked to a variety of biological functions. Cinnamon essential oil, eugenol, has a higher antimicrobial action than other essential oils [19]. The cinnamon ethanolic extract has wound-healing properties [20]. Increased superoxide dismutase (SOD) and collagen deposition are two indicators that natural bioactive compounds improve tissue regeneration through the modulation of oxidative stress, according to research using fisetin, a key flavonoid [21]. Additionally, a recent study utilizing electrospun nanofiber formulations enhanced with plant extracts highlights that the gold standard for successful wound treatment combines strong antibacterial activity with improved tissue regeneration (re-epithelialization and collagen synthesis) [22]. Flavonoids and polyphenols are abundant in both M. pudica and C. verum. This study provides a mechanistic basis for the hypothesis that the extracts' effectiveness is partially mediated by scavenging free radicals (raising SOD/TAC) and reducing oxidative stress (lowering MDA/TOS).

Bangladeshi rural farmers have long used a variety of plant and herbal medicines, as well as inorganic materials like cotton ash, for the healing of skin lesions [23].

Marigold leaves, turmeric, aloe vera, and ashes have all been used to treat the wounds of goats, lambs, and rabbits [24]. The therapeutic efficacies of Lajjabati (*Mimosa pudica*) and Daruchini (*Cinnamomum verum*) were not contrasted with those of the other herbal plants, though.

Although there is plenty of traditional knowledge, the application frequently lacks a scientifically supported foundation for choosing one treatment over another for the best chance of recovery. In particular, neither Lajjabati (*Mimosa pudica*) nor Daruchini's (*Cinnamomum verum*) medicinal benefits were directly compared to those of the other herbal plants or to one another in a controlled in vivo setting in order to determine which provides better healing promotion. A comparison study of this type is essential because histological analysis can provide important, micro-level proof of tissue repair, such as the degree of re-epithelialization and the caliber of collagen deposition, which clinical observations alone are unable to detect. A direct comparative analysis using a standardized wound model, such as the incised surgical wound, is required to validate and optimize traditional veterinary medicine practices because of the strong anecdotal evidence supporting the wound-healing qualities of both *Mimosa pudica* and *Cinnamomum verum*. Therefore, this study was carried out to explore the therapeutic efficacies of Lajjabati (*Mimosa pudica*) leaves and Daruchini (*Cinnamomum verum*) paste on the healing of incised surgical wounds and to observe the histo-architectural changes of wounded skin in rabbits.

## Materials and methods

The study was carried out from October 2022 to March 2023 at Bangladesh Agricultural University (BAU), Mymensingh, in the Department of Surgery and Obstetrics.

### Ethical approval

The Animal Welfare and Experimentation Ethics Committee (AWEEC) of Bangladesh Agricultural University (BAU), Mymensingh, created rules for the care and use of animals, which were followed in all experimental methods [(Approval Number: AWEEC/ BAU/2023 (29)].

### Experimental animals

For this experiment, sixteen rabbits that seemed to be in good health were used. The rabbits were purchased from the local rabbit farm of Muktagacha, Mymensingh. The animals' body weights varied from 1.5 to 2.0 kg. The animals were housed in standard laboratory settings with veterinary monitoring and no limitations on water or food. The rabbits were kept in quarantine for two weeks before the experiment. Following the experimental period, rabbits were maintained in the rabbit shed within the Department of Surgery and Obstetrics. They received standardized nutritional provisions and supplementation throughout the entire recovery phase, after which they were designated for ongoing research protocols.

### Experimental design

Sixteen rabbits were randomly assigned to four experimental groups (n = 4) using a simple lottery system to ensure unbiased treatment allocation. All wound measurements and histological assessments were performed by investigators who were blinded to the treatment groups to minimize observational bias.

Thirty-two (32) surgical incisions were created on the skin of sixteen rabbits, with two in each. Four treatment groups of rabbits were formed, i.e., Groups A, B, C, and D each contain four animals. Fresh Lajjabati *(Mimosa pudica)* and Daruchini (*Cinnamomum verum*) paste were applied daily to the skin wounds of animals in Group A and Group B, respectively. In Group C, a mixture of Lajjabati (*Mimosa pudica*) and Daruchini (*Cinnamomum verum*) paste was applied to skin wounds. To prevent tampering with the production of granulation tissue, these animals were kept under close observation. Normal saline was used to bathe the wounds in Group D, which was kept as a control.

## Preparation of Lajjabati leave and Daruchini paste

Fresh Lajjabati (*Mimosa pudica*) plants and Daruchini (*Cinnamomum verum*) were collected from the Agronomy field station and local market of Bangladesh Agricultural University, Mymensingh, respectively, and washed carefully with clean water. Lajjabati (*Mimosa pudica*) leaves and Daruchini (*Cinnamomum verum*) were peeled and cleaned again with water. The cleaned lajjabati (*Mimosa pudica*) leaves and daruchini (*Cinnamomum verum*) were cut into small slices, and then each was separately made into a paste using a mortar and pestle. To make the paste, additional water was not added; it was just leftover water from rising. Finally, the prepared paste was applied directly onto the skin wound.

## Wound creation and closure

Wounds were created on the flank region of rabbits. The animals were anesthetized locally by subcutaneous infiltration of 2% Lidocaine Hydrochloride (Jasocaine®, Jayson Pharmaceuticals, Bangladesh) at the incision site. The rabbits were positioned on lateral recumbency. The operation sites were prepared aseptically. Surgical incisions were made vertically in the flank region of the rabbit to create the wounds. The length and depth of each incision were 2.5 cm and 0.5 cm, respectively. Following the incision, blunt dissection was performed to separate the skin from the underlying tissues. Simple interrupted sutures were used to close the wounds using nylon. Every suture was positioned 8 mm apart, and there was a 5 mm gap between the needle's location and the cutting edge's border.

## Local treatment

Following the suture, Lajjabati (*Mimosa pudica*) paste, Daruchini (*Cinnamomum verum*) paste, and their mixture were used to treat the skin wounds of different groups. Systemic antibiotics, antihistamines, or anti-inflammatory drugs were avoided to overcome their potential impact on wound healing. Food was not denied, and a highly digestible meal was provided within 24 hours of surgery.

## Observation of morphological characters

From Day 0 to Day 21 following surgery, follow-up data were gathered. After 7 days of the operation, all the sutures were removed. The wound healing process was assessed by documenting certain morphological features such as the swelling area of the wound, the elevation of the sutured line from the skin surface, and the width of the sutured area. On Day 7 of the procedure, the elevation of the sutured line was observed during suture removal. The width of the sutured area was measured on Day 0, Day 7, Day 14, and Day 21 to determine wound contraction length. To assess the effects of the treatments on wound healing, the swelling area of the wound, elevation of the sutured line, the length of wound contraction, and the width of the sutured area were measured using slide calipers. Every day, all wounds were carefully observed to look for any complications.

## Assessment of wound healing

Every day, the healing process was observed in each group of animals. The healing score was categorized as Excellent, Good, and Fair. Excellent healing resulted in no inflammation, exudation, infection, dehiscence, and a progressive reduction in the cutting edge's width. Good: minimal exudation and inflammation, no dehiscence, progressive narrowing of the cutting edge; Fair: significant inflammation, infection, and exudation [25].

## Bacteriological study

For the bacteriological study, swabs were collected aseptically from the skin wounds of all treatment groups from Day 1 to Day 8. The gathered samples were kept overnight at 37°C in the nutrient broth. After that, the samples were inoculated using the streak plate method in nutrient agar and incubated overnight at 37°C. To observe the morphological and staining characteristics of bacteria, Gram staining was performed.

## Histopathology

On Day 3, Day 7, and Day 14 following the surgery, skin samples were taken from the wound areas of every animal in every treatment group. For histopathology, the tissues with dermis and epidermis were preserved in 10% buffered neutral formalin solution for more than seven days. The fixed tissue samples were trimmed into 1.5x1 cm, washed in running tap water, dehydrated by ascending ethanol series, cleaned in chloroform for 3 hours, and impregnated in melted paraffin (56–60°C). The tissues were sectioned (5 μm thickness) with a microtome and stained with routine hematoxylin and eosin stain as per standard protocol [26] at the laboratory, Department of Surgery and Obstetrics, Bangladesh Agricultural University (BAU), Mymensingh. The stained slides were observed under an Olympus photomicroscope (CX43), and photographs of these slides were taken at the Department of Physiology, BAU, Mymensingh.

To objectively evaluate tissue repair, a semi-quantitative histological scoring system was applied. Parameters, including inflammation, fibroblast proliferation, and glandular change, were graded on a scale of 0–3 (0: Absent; 1: Mild/Scanty; 2: Moderate; 3: Severe/Robust) (Table 1).

## Statistical analysis

Statistical analysis was performed using IBM SPSS Statistics (version 22). Results are expressed as Mean±SEM (Standard Error of Mean). A one-way analysis of variance (ANOVA) was conducted to compare data among the different groups. Subsequent pairwise comparisons were performed using Tukey's HSD post-hoc test to ascertain differences in individual parameters among the experimental groups. Statistical significance was determined at a P-value of less than 0.05.

## Results

### Morphological characteristics of skin wounds

Morphological characteristics such as swelling of the sutured area, elevation of the sutured line, wound contraction, and healing score are presented in Table 2 and S1 Table. The swelling was observed up to three days post-operation, as it decreased from Day 3. The swelling areas of wounds did not significantly vary among the groups. However, there was little difference between Group A (11.69±0.21 mm) and Group B (11.50±0.13 mm) (Daruchini (*Cinnamomum verum*)-treated group) in terms of the swelled area of wounds. Morphological assessment (Table 2) indicated that Group

**Table 1.** Histological scoring of different wound healing parameters.

| Parameter | Score 0 | Score 1 (Mild) | Score 2 (Moderate) | Score 3 (Severe) |
|---|---|---|---|---|
| Inflammation | None | Few cells | Moderate aggregates | Dense infiltration |
| Fibrosis | None | Thin/Faint fibers | Clearly visible | Thick bundles |
| Glandular Change | Normal | Slight enlargement | Hyperplasia | Marked hyperplasia |

**Table 2.** The effects of Lajjabati (*Mimosa pudica*) paste (Group A), Daruchini (*Cinnamomum verum*) paste (Group B), their mixture (Group C), and normal saline (Group D) on wound healing in rabbits (Mean±SEM).

| Groups | Swelling of sutured area (mm) | Elevation of sutured line (mm) | Contraction length (mm) | Healing Score |
|---|---|---|---|---|
| Group A | 11.69±0.21 | 3.50±0.13[a] | 1.81±0.16 | Excellent |
| Group B | 11.50±0.13 | 2.54±0.10[b] | 1.93±0.15 | Excellent |
| Group C | 12.19±0.25 | 3.56±0.12[a] | 1.88±0.13 | Good |
| Group D | 12.19±0.25 | 3.56±0.15[a] | 1.81±0.13 | Fair |

a, b in the same column indicate a significant (p<0.05) difference among the groups

B exhibited a significantly lower (P<0.05) elevation of the suture line (2.54±0.10 mm) compared to Group A and the control. However, wound contraction rates did not show significant differences between the herbal treatment groups and the control. The groups treated with lajjabati (*Mimosa pudica*) and daruchini (*Cinnamomum verum*) (Group A and Group B, respectively) had excellent wound healing scores. Group C, when the skin wound was treated with a combination of lajjabati (*Mimosa pudica*) and daruchini (*Cinnamomum verum*) had good wound healing scores (Table 2). From Day 0 to Day 21 postoperatively, the sutured area's width was measured to monitor the wound healing process. While initial wound widths on Day 0 showed slight variation between groups, the rate of contraction was analyzed relative to the starting baseline. Regardless of treatment, the width of the sutured area grew until Day 3 and progressively declined in all wounds (Table 3). The width of the sutured area was significantly different at Day 0, Day 3, and Day 14 among the groups (S2 Table). However, on Days 14 and 21, the wound surface area of the treated lesions decreased compared to the area on Days 0–7.

### Identification of bacteria

The presence of bacterial colonies in nutrient agar is shown in Fig 1. The presence of Staphylococcus species, which showed gray white to yellowish colonies on nutrient agar, was confirmed by the results of the Gram's staining.

### Bacterial colony count

A total bacterial colony count per day in wounds of different groups is presented in Fig 2. Microbiological studies showed that bacterial colonies were present in the culture of samples of the treated group from Day 1–8. The bacterial colony decreases gradually day by day among the treated groups. The difference in mean values varies significantly (P<0.05) among the groups. The bacterial colony count was significantly lower compared to other groups (S1 Appendix).

### Histopathological examination

Semi-quantitative histological scoring (Table 4) confirmed that Group A (Lajjabati) demonstrates the most efficient recovery, transitioning from peak scores of 3 in inflammation and fibroblast proliferation at Day 3 to a complete resolution (score 0) across all parameters by Day 14. In contrast, Group B (Daruchini) and Group C (Mixed) showed significantly higher scores (score 3) for glandular hyperplasia and persistent inflammation, respectively, which correlates with the tissue irritation observed during gross morphological assessment.

Histopathological examinations were performed on Day 3, Day 7, and Day 14, and histoarchitectural study showed some variations in their action in wound healing. The results are shown in Figs 3, 4, and 5, respectively.

**Histopathological observations on Day 3.** Presence of inflammation and swelling of subcutaneous tissue in Group A when the wound was treated by lajjabati (*Mimosa pudica*) (Fig 3A-yellow circle). Presence of less inflammation,

**Table 3. Width (mm) of the sutured area of wounds treated with lajjabati (*Mimosa pudica*) (Group A), daruchini (*Cinnamomum verum*) (Group B), a mixture of lajjabati (*Mimosa pudica*) and daruchini (*Cinnamomum verum*) (Group C), and normal saline (Group D) on wound healing in rabbits (Mean± SEM).**

| Groups | Width of sutured area (mm) | | | | |
|---|---|---|---|---|---|
| | Day 0 | Day 3 | Day 7 | Day 14 | Day 21 |
| Group A | 7.50±0.13[a] | 8.06±0.11[a] | 7.69±0.19 | 7.00±0.13[a] | 5.06±0.15 |
| Group B | 6.44±0.15[b] | 7.32±0.20[b] | 7.00±0.19 | 6.13±0.12[b] | 4.69±0.16 |
| Group C | 7.50±0.13[a] | 8.31±0.10[a] | 7.69±0.28 | 7.13±0.13[a] | 5.13±0.13 |
| Group D | 7.50±0.13[a] | 8.37±0.23[a] | 7.69±0.27 | 7.00±0.13[a] | 5.06±0.15 |

a, b in the same column indicate significant (p<0.05) differences among the groups

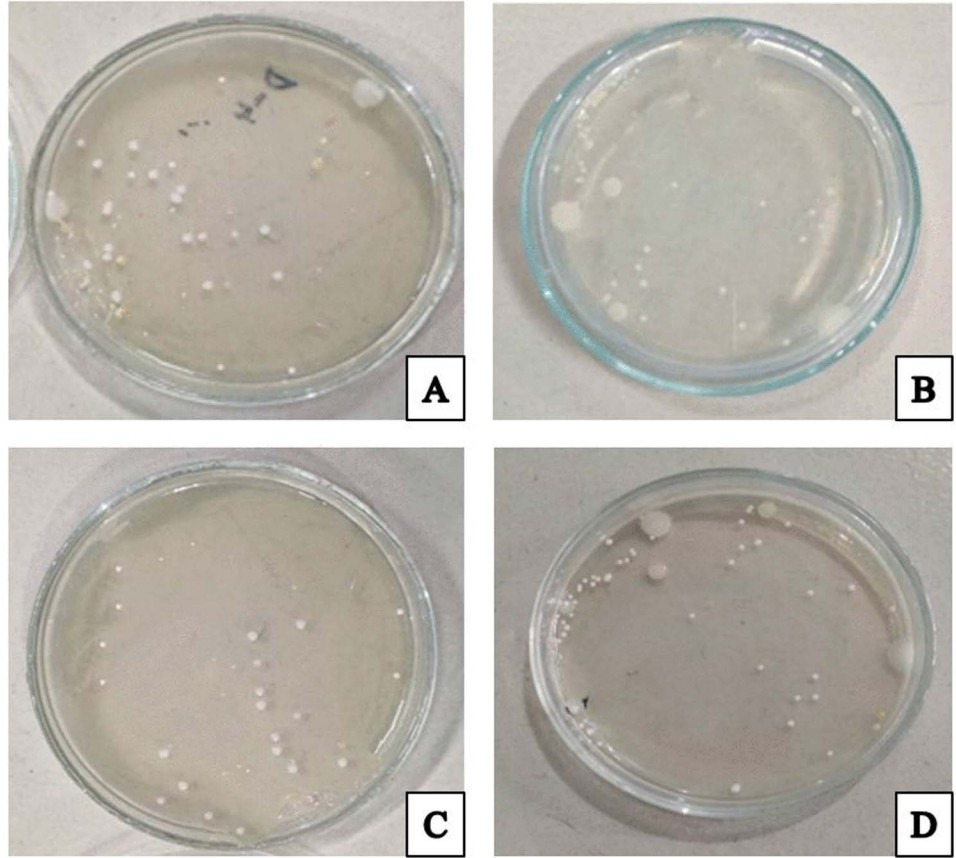

**Fig 1. Presence of bacterial colony in primary culture of nutrient agar observed in different samples collected from Group A (lajjabati paste) (A), Group B (daruchini paste) (B), Group C (Mixed) (C), and Group D (Control) (D).**

hyperplasia of the gland, fibrosis, and swelling of subcutaneous tissue in the daruchini (*Cinnamomum verum*)-treated Group (Fig 3B). The presence of inflammation and marked proliferation of fibrous tissue (yellow circle) in Group C, while the mixture of lajjabati (*Mimosa pudica*) and daruchini (*Cinnamomum verum*) was applied (Fig 3C). Presence of abundant inflammation and no swelling in the epidermal tissue in Group D (Fig 3D-Arrow)

**Histopathological observations on Day 7.** Reduction of inflammation and growth of hair follicles appear in the wound of the lajjabati (*Mimosa pudica*)-treated group (Fig 4A- yellow circle). Histopathological study revealed a little bit of reduction of Inflammation, and hypoplastic glands were present in Group B when daruchini (*Cinnamomum verum*) was applied in the skin wound (Fig 4B- yellow circle). Inflammation was present, and proliferation of fibrous tissue occurred in Group C while it was treated by the mixture of the lajjabati (*Mimosa pudica*) and daruchini (*Cinnamomum verum*) paste (Fig 4C). Presence of less inflammation in Group D (Fig 4D).

**Histopathological observations on Day 14.** Histoarchitectural study showed that all the structures of the skin appeared normal, and the presence of hair follicles in Group A, the wound treated by lajjabati (*Mimosa pudica*) paste (Fig 5A-Star). The proliferation of fibrous tissue occurred in Group B while the daruchini (*Cinnamomum verum*) paste was applied in the wound (Fig 5B- yellow circle). The study revealed the presence of inflammation and hypoplastic gland in Group C when the mixture of both pastes was supplied in the wounded area (Fig 5C-yellow circle). The growth of hair follicles and their structure appear normal in Group D (Fig 5D).

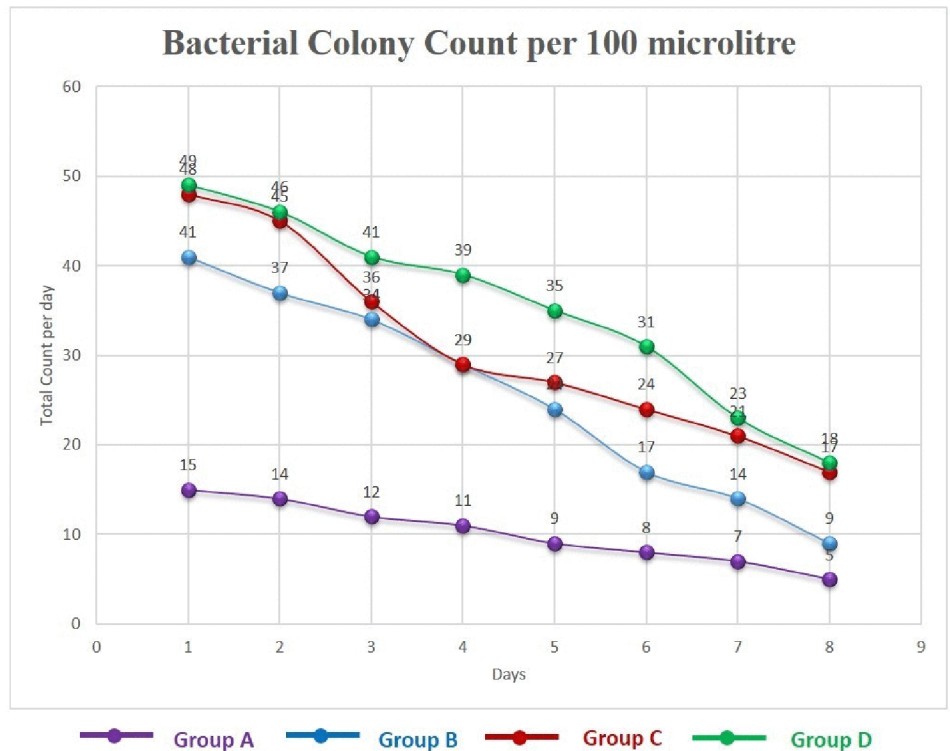

**Fig 2. Bacterial colony count (per 100 microlitre) in the samples collected from Group A (lajjabati paste) (A), Group B (daruchini paste) (B), Group C (Mixed) (C), and Group D (Control) (D) from Day 1 to Day 8.**

**Table 4. Histopathological scoring of wound healing parameters at different days of wound.**

| Groups | Inflammation | | | Fibroblast proliferation | | | Glandular hyperplasia | | |
|---|---|---|---|---|---|---|---|---|---|
| | Day 3 | Day 7 | Day 14 | Day 3 | Day 7 | Day 14 | Day 3 | Day 7 | Day 14 |
| Group A | 3 | 0 | 0 | 3 | 1 | 1 | 2 | 1 | 0 |
| Group B | 2 | 1 | 0 | 3 | 2 | 2 | 3 | 3 | 3 |
| Group C | 1 | 1 | 1 | 2 | 2 | 2 | 2 | 1 | 3 |
| Group D | 3 | 1 | 0 | 2 | 2 | 1 | 1 | ±2 | 1 |

Scores are based on a scale of 0–3 (0 = Absent, 1 = Mild, 2 = Moderate, 3 = Marked/Robust)

## Discussion

Traditional wound treatment practices use a variety of plant and herbal medicines to speed up the healing process. In Bangladesh, a large variety of herbal plants are available. However, they are not popularly used for expediting the healing process, and their efficacy for wound healing has not been determined. So, this experiment was designed to investigate and compare the effects of Lajjabati (*Mimosa pudica*), Daruchini (*Cinnamomum verum*), and a combination of both products on the healing of surgical wounds in rabbits.

To restore normal structural and functional integrity in wound healing, various cellular activities interact step by step. The inflammatory phase is the initial stage of wound healing. An incision that goes through the entire thickness of the skin disrupts the microvasculature and results in instantaneous bleeding. After the bleeding has been stopped, inflammatory

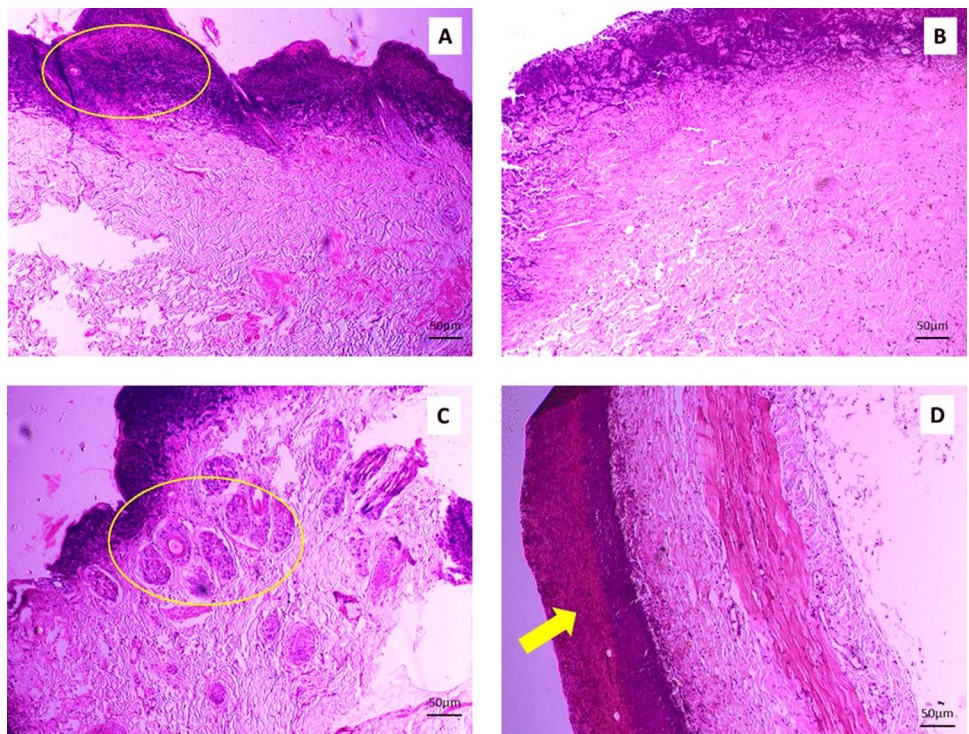

**Fig 3. Histopathological observations of the healing process of skin wound stained with H and E (10x).** Tissue samples were collected from Group A (Lajjabati paste) **(A)**, Group B (Daruchini paste) **(B)**, Group C (Mixed) **(C)**, and Group D (Control) (D) on Day 3. The study revealed the presence of inflammation and swelling of subcutaneous tissue in Group A (3A-yellow circle), marked proliferation of fibrous tissue in Group C (3C-yellow circle), and the presence of abundant inflammation and no swelling in the epidermal tissue in Group D (3D-yellow arrow).

cells move (chemotaxis) into the wound to initiate the inflammatory phase, which is marked by the successive infiltration of lymphocytes, neutrophils, and macrophages [27]. Up to Day 21 postoperatively, the increased swelling following treatments was observed in this study. The width of the sutured area was also measured from Day 0 to Day 21 postoperatively to know how the wound contracted anatomically during the healing process. From Day 14, the width of the sutured line significantly decreased in all treated groups. This is corroborated by Slatter, 2002 [28], who stated that the maximum time after an accident for wound closure is 5–15 days. Additionally, there was no discernible difference between the wounds of the four groups in terms of contraction length per week. This result validates the hypothesis that the myofibroblast, located at the margin of the wound and associated with extracellular matrix components and myofibroblast development, is necessary for wound contraction [29].

In comparison to the other three groups, Group D (control) wounds treated with normal saline showed the highest swelling area and elevation of the sutured line. Interestingly, while Group B (*Cinnamomum verum*) exhibited the lowest elevation of the sutured line, histopathological analysis revealed marked tissue irritation and glandular hyperplasia. This suggests that the lower morphological elevation was not an indicator of superior healing, but rather a different tissue response to the irritant properties of *Cinnamomum verum*. In contrast, Group A (*Mimosa pudica*) demonstrated a balanced progression through the inflammatory and remodeling phases, as evidenced by the early appearance of hair follicles. The wound contraction length treated with Lajjabati (*Mimosa pudica*) paste, Daruchini (*Cinnamomum verum*) paste, and their mixed paste did not show any noticeable differences.

The general histological abnormalities were identified as increasing fibrillar tissue, edema, and variable degrees of inflammatory cell infiltration. The histological analysis of all groups showed infiltration of the inflammatory cells. On Day

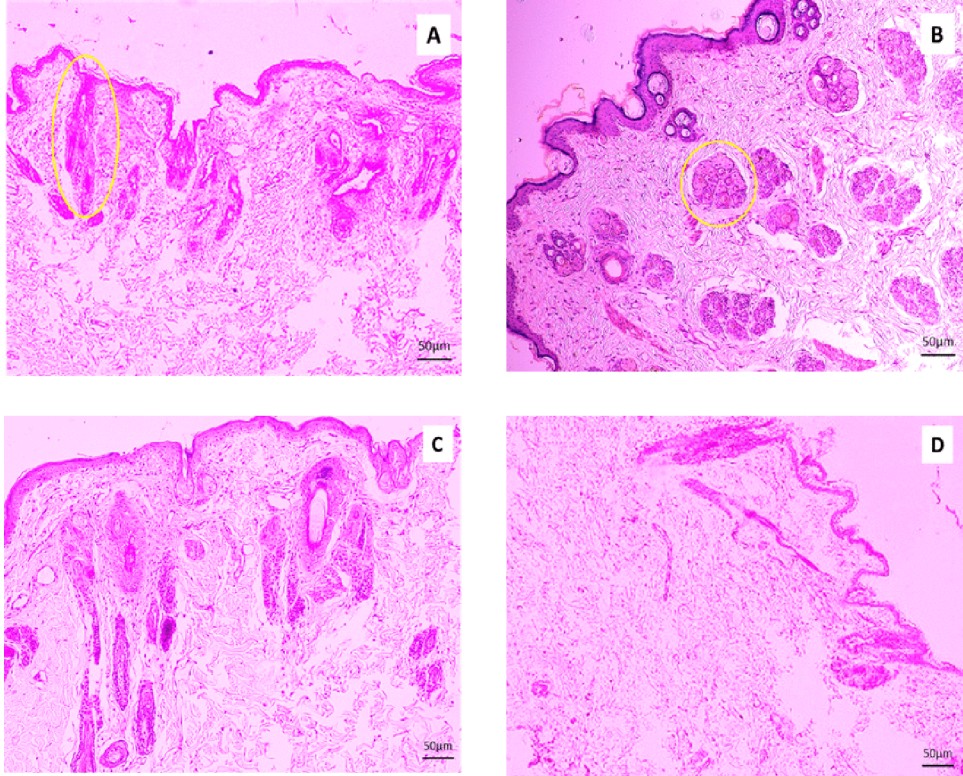

**Fig 4. Histopathological evaluation of the section of skin wound stained with H and E (10x).** Tissue samples were collected from Group A (Lajjabati paste) **(A)**, Group B (Daruchini paste) **(B)**, Group C (Mixed) **(C)**, and Group D (Control) **(D)** on Day 7. Reduction of inflammation and growth of hair follicles appear in the wound of lajjabati-treated group (4A- yellow circle). Histopathological study revealed a little bit of reduction of Inflammation, and hypoplastic glands were present in Group B when daruchini was applied in the skin wound (4B- yellow circle).

3, Group A had the most inflammation based on cellular proliferation, followed by Group B and Group C. The changes in sample times and treatment approaches used for the experimental groups were the cause of the variations in the corresponding histopathological characteristics. More inflammatory cell infiltration suggests that the inflammatory phase and early wound healing events are being promoted.

In this study, enhanced wound healing efficacy was observed in the wound treated with Lajjabati (*Mimosa pudica*) paste compared to Daruchini (*Cinnamomum verum*) paste and their mixture. *Mimosa pudica* root extract has also been found to significantly increase the tensile strength of incised wounds [13,16] which supports the result. Moreover, better wound healing efficacy in the Lajjabati (*Mimosa pudica*)-treated group might be due to the antimicrobial and antioxidant properties of Lajjabati (*Mimosa pudica*) [13]. Many studies reveal that turmeric, honey, and aloe vera are the substances that are thought to be beneficial for promoting wound healing by inhibiting platelet aggregation, inflammatory cytokine release, oxidative stress, metastasis, and epithelization [23,30,31]. Aloe vera and honey dressings performed better than silver sulfadiazine, according to several studies [32–34], with significantly faster healing times and improved sterilization of infected wounds.

Microbiological analysis of the samples used in this study indicated the presence of bacterial colonies. The group treated with Lajjabati (*Mimosa pudica*) was found to have the lowest microorganisms, followed by the group treated with Daruchini (*Cinnamomum verum*) and a combination of these two. According to this research, Lajjabati (*Mimosa pudica*) extract has higher antibacterial effects on wounds, which helps to speed up wound healing and avoid postoperative

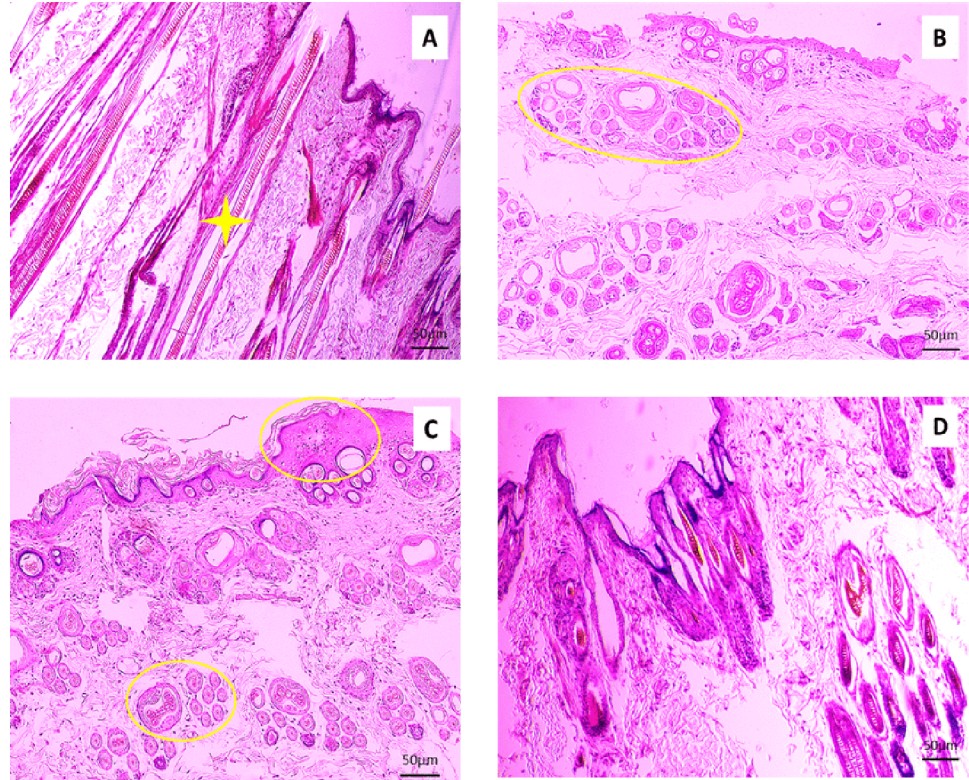

**Fig 5. Histopathological analysis of the section of skin wound stained with H and E (10x).** Tissue samples were collected from Group A (Lajjabati paste) **(A)**, Group B (Daruchini paste) **(B)**, Group C (Mixed) **(C)**, and Group D (Control) **(D)** on Day 14. Histoarchitectural study showed that all the structures of the skin appeared normal, and the presence of hair follicles in Group A, the wound treated by lajjabati paste (5A-yellow star). Proliferation of fibrous tissue occurred in Group B while the daruchini paste was applied in the wound (5B- yellow circle). The study revealed the presence of inflammation and hypoplastic glands in Group C when the mixture of each paste was supplied in the wounded area (5C-yellow circle).

infections. In our investigation, the presence of gram-positive bacteria, specifically Staphylococcus species, was confirmed by Gram staining. It found that Staphylococcus aureus responded better to Lajjabati (*Mimosa pudica*) extract [13]. It is supported by another author who said, the chemical components of Lajjabati (*Mimosa pudica*) can more easily enter the bacterial cell and kill the pathogen [35].

The positive findings of this study of Lajjabati (*Mimosa pudica*) have important practical applications. Because it is a cheap and plentiful plant resource, its extract has the potential to be an accessible and affordable medicinal agent, especially in areas where conventional pharmaceutical treatments are limited. On the other hand, the results about Daruchini (*Cinnamomum verum*), which showed the ability to hinder healing when used concurrently, underscore the critical need for pre-clinical validation of herbal combinations. This demonstrates that mixing natural products is not always advantageous and necessitates cautious screening to avoid unintended adverse tissue reactions.

While this study provides valuable preliminary data on the efficacy of herbal extracts, it is constrained by using an animal model, a lack of functional biomechanical assessments such as tensile strength testing, and a relatively small sample size (n = 4 per group). In addition to the comprehensive characterization and isolation of the active components in Lajjabati (*Mimosa pudica*) and further exploration of the Daruchini (*Cinnamomum verum*) effect, subsequent research should prioritize validation in additional in vivo models or clinical situations. Furthermore, more research is necessary to completely define the functional quality of tissue restoration, and this research will include conclusive biochemical and biomechanical

endpoints. In particular, tensile strength testing to evaluate the structural and functional integrity of the repaired tissue and measuring hydroxyproline content as a direct proxy for collagen deposition will offer a more thorough understanding of the intervention's long-term efficacy and remodeling potential.

## Conclusion

According to the gross appearance and morphological characteristics of the wound, both Lajjabati (*Mimosa pudica*) and Daruchini (*Cinnamomum verum*) pastes initially appear effective and contribute to cutaneous wound healing. However, the detailed histo-architectural changes revealed critical differences in the quality of tissue regeneration. Specifically, Daruchini (*Cinnamomum verum*) application resulted in a lower elevation of the suture line, which was associated with greater tissue reactions, including hyperplasia, enlargement of skin glands, fibrosis, and prolonged inflammation, suggesting that its topical application in this concentration may induce irritant effects that could potentially delay the remodeling phase. On the other hand, by the seventh day after the wound, Lajjabati (*Mimosa pudica*) paste showed significantly reduced inflammation and the early regeneration of hair follicles and other skin structures, reinforcing its superior tissue restoration. In conclusion, Lajjabati (*Mimosa pudica*) extract shows promising therapeutic potential as an accessible, low-cost wound-healing agent, particularly in resource-limited settings. However, given the small sample size and the specific focus on an acute surgical model, these findings should be considered preliminary. Further research involving larger cohorts, biomechanical strength testing, and biochemical analysis of collagen deposition is required before these herbal treatments can be broadly recommended for clinical use in human or veterinary medicine.

## Supporting information

**S1 Table. Statistical comparison (p-values and confidence intervals) for the area of swelling and elevation of the wound.**
(DOCX)

**S2 Table. Statistical comparison (p-values and confidence intervals) for the Width (mm) of the sutured area of wounds at Day-0 to day-21 of different groups.**
(DOCX)

**S1 Appendix. Statistical comparison of bacterial colony count with the statistical differences among different groups (Mean, SE and 95% confidence interval).**
(DOCX)

**S1 File. Minimal data set.**
(DOCX)

## Acknowledgments

We would like to acknowledge the Department of Surgery and Obstetrics and the Department of Physiology for providing laboratory support. The Bangladesh Agricultural University Research System (BAURES), which provided all kinds of technical support, is also acknowledged by the authors.

## Author contributions

**Conceptualization:** Rukhsana Amin Runa.

**Data curation:** Md. Abdur Rahim Prodhan, Suravi Akter.

**Formal analysis:** Rukhsana Amin Runa, Md. Abdur Rahim Prodhan, Afrina Mustari.

**Funding acquisition:** Md. Abdur Rahim Prodhan.

**Investigation:** Rukhsana Amin Runa, Md. Abdur Rahim Prodhan.

**Methodology:** Rukhsana Amin Runa, Md. Abdur Rahim Prodhan.

**Project administration:** Rukhsana Amin Runa.

**Resources:** Rukhsana Amin Runa, Afrina Mustari.

**Supervision:** Rukhsana Amin Runa.

**Validation:** Rukhsana Amin Runa.

**Visualization:** Rukhsana Amin Runa, Md. Abdur Rahim Prodhan, Suravi Akter, Afrina Mustari.

**Writing – original draft:** Md. Abdur Rahim Prodhan.

**Writing – review & editing:** Rukhsana Amin Runa, Suravi Akter, Afrina Mustari.

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
