## [Decision Letter · Decision Letter 0]

5 Oct 2025

PONE-D-25-41676Efficacy of Lajjabati (Mimosa pudica) and Daruchini (Cinnamomum verum) extracts on wound healing in rabbitsPLOS ONE

Dear Dr. Runa,

Thank you for submitting your manuscript to PLOS ONE. After careful consideration, we feel that it has merit but does not fully meet PLOS ONE’s publication criteria as it currently stands. Therefore, we invite you to submit a revised version of the manuscript that addresses the points raised during the review process.

We look forward to receiving your revised manuscript.

Kind regards,

Vinod Ayyappan, Ph.D., PostDoc

Academic Editor

PLOS ONE

**Journal Requirements:**

1. When submitting your revision, we need you to address these additional requirements. Please ensure that your manuscript meets PLOS ONE's style requirements, including those for file naming. The PLOS ONE style templates can be found at https://journals.plos.org/plosone/s/file?id=wjVg/PLOSOne_formatting_sample_main_body.pdf and https://journals.plos.org/plosone/s/file?id=ba62/PLOSOne_formatting_sample_title_authors_affiliations.pdf 2. In your Methods section, please include a comment about the state of the animals following this research. Were they euthanized or housed for use in further research? If any animals were sacrificed by the authors, please include the method of euthanasia and describe any efforts that were undertaken to reduce animal suffering. 3. Thank you for stating the following financial disclosure: The authors appreciate the financial support received from the National Science and Technology (NST), Ministry of Science and Technology (MoST), Government of the People’s Republic of Bangladesh.    Please state what role the funders took in the study.  If the funders had no role, please state: "The funders had no role in study design, data collection and analysis, decision to publish, or preparation of the manuscript." If this statement is not correct you must amend it as needed. Please include this amended Role of Funder statement in your cover letter; we will change the online submission form on your behalf. 4. We note that your Data Availability Statement is currently as follows: All relevant data are within the manuscript and its Supporting Information files. Please confirm at this time whether or not your submission contains all raw data required to replicate the results of your study. Authors must share the “minimal data set” for their submission. PLOS defines the minimal data set to consist of the data required to replicate all study findings reported in the article, as well as related metadata and methods (https://journals.plos.org/plosone/s/data-availability#loc-minimal-data-set-definition). For example, authors should submit the following data: - The values behind the means, standard deviations and other measures reported;- The values used to build graphs;- The points extracted from images for analysis. Authors do not need to submit their entire data set if only a portion of the data was used in the reported study. If your submission does not contain these data, please either upload them as Supporting Information files or deposit them to a stable, public repository and provide us with the relevant URLs, DOIs, or accession numbers. For a list of recommended repositories, please see https://journals.plos.org/plosone/s/recommended-repositories. If there are ethical or legal restrictions on sharing a de-identified data set, please explain them in detail (e.g., data contain potentially sensitive information, data are owned by a third-party organization, etc.) and who has imposed them (e.g., an ethics committee). Please also provide contact information for a data access committee, ethics committee, or other institutional body to which data requests may be sent. If data are owned by a third party, please indicate how others may request data access. 5. If the reviewer comments include a recommendation to cite specific previously published works, please review and evaluate these publications to determine whether they are relevant and should be cited. There is no requirement to cite these works unless the editor has indicated otherwise.

**Additional Editor Comments:**

Reviewer #1:

The manuscript presents an interesting and relevant study on the comparative efficacy of M. pudica (Lajjabati) and C. verum (Daruchini) extracts on wound healing in rabbits. The topic is important for ethnomedicine and could contribute useful data to the field of herbal wound management. The experimental design is generally sound, and the results are clearly presented with supportive histopathological and bacteriological data. However, some aspects of the manuscript require clarification, additional methodological details, and improvements in presentation to improve the rigor and clarity of the study. This section could be strengthened by citing following literature to better contextualize the study:

-https://doi.org/10.30500/ivsa.2025.519077.1444

-https://doi.org/10.30500/ivsa.2024.428433.1382

Introduction

The introduction provides a good overview of wound healing and traditional herbal treatments. It would benefit from more details on the rationale for choosing these two plants, especially the expected synergistic or differential effects. Some recent references on herbal wound healing could be included to strengthen the background.

Results

The morphological, bacteriological, and histopathological findings are clearly described. The graphical figures related to bacteria colony counts could be supplemented with statistical comparisons to highlight significant differences. The histopathology images support the text well, although higher resolution images with scale bars would improve visual interpretation.

Discussion and Conclusion

The discussion appropriately relates findings to previous research. The statement regarding Daruchini causing delayed healing is interesting but should avoid overgeneralization; it might be due to irritation or tissue reactions, but further mechanistic studies are warranted. Consider emphasizing the practical implications and potential limitations, such as small sample size or animal model limitations. Suggest potential future directions for validation in other models or clinical settings.

The conclusion is consistent with the results and highlights the potential of Lajjabati extract as a wound healing agent. Slightly soften conclusions about Daruchini, as “antagonist” effects require more evidence.

Overall, the manuscript addresses a valuable research question with adequate methodology and supporting data. Addressing the above comments will strengthen the clarity, reproducibility, and impact of the study

Reviewers' comments:

Reviewer's Responses to Questions

**Comments to the Author**

1. Is the manuscript technically sound, and do the data support the conclusions?

Reviewer #1: Yes

Reviewer #2: Partly

2. Has the statistical analysis been performed appropriately and rigorously? 

Reviewer #1: Yes

Reviewer #2: No

3. Have the authors made all data underlying the findings in their manuscript fully available?

Reviewer #1: Yes

Reviewer #2: Yes

4. Is the manuscript presented in an intelligible fashion and written in standard English?

Reviewer #1: Yes

Reviewer #2: Yes

5. Review Comments to the Author

**Reviewer #1:** The manuscript presents an interesting and relevant study on the comparative efficacy of M. pudica (Lajjabati) and C. verum (Daruchini) extracts on wound healing in rabbits. The topic is important for ethnomedicine and could contribute useful data to the field of herbal wound management. The experimental design is generally sound, and the results are clearly presented with supportive histopathological and bacteriological data. However, some aspects of the manuscript require clarification, additional methodological details, and improvements in presentation to improve the rigor and clarity of the study. This section could be strengthened by citing following literature to better contextualize the study:

-https://doi.org/10.30500/ivsa.2025.519077.1444

-https://doi.org/10.30500/ivsa.2024.428433.1382

Introduction

The introduction provides a good overview of wound healing and traditional herbal treatments. It would benefit from more details on the rationale for choosing these two plants, especially the expected synergistic or differential effects. Some recent references on herbal wound healing could be included to strengthen the background.

Results

The morphological, bacteriological, and histopathological findings are clearly described. The graphical figures related to bacteria colony counts could be supplemented with statistical comparisons to highlight significant differences. The histopathology images support the text well, although higher resolution images with scale bars would improve visual interpretation.

Discussion and Conclusion

The discussion appropriately relates findings to previous research. The statement regarding Daruchini causing delayed healing is interesting but should avoid overgeneralization; it might be due to irritation or tissue reactions, but further mechanistic studies are warranted. Consider emphasizing the practical implications and potential limitations, such as small sample size or animal model limitations. Suggest potential future directions for validation in other models or clinical settings.

The conclusion is consistent with the results and highlights the potential of Lajjabati extract as a wound healing agent. Slightly soften conclusions about Daruchini, as “antagonist” effects require more evidence.

Overall, the manuscript addresses a valuable research question with adequate methodology and supporting data. Addressing the above comments will strengthen the clarity, reproducibility, and impact of the study

**Reviewer #2:**

This manuscript explores the efficacy of Mimosa pudica and Cinnamomum verum extracts in wound healing in rabbits. The topic is relevant, and the experimental design (four groups, multiple endpoints, histology, and microbiology) is appropriate. The results suggest that Mimosa pudica has superior wound healing properties, while Cinnamomum verum may delay healing when used alone or in combination. However, the study has methodological weaknesses that need to be addressed before it can be considered for publication.

Major Comments

1. Sample size: Only four animals per group were used. Please justify the sample size with a power analysis or acknowledge this as a limitation.

2. Statistical analysis: Provide exact p-values and confidence intervals for all comparisons. Clarify group differences in tables. Healing scores should include inter-rater reliability or a validated scoring system.

3. Histopathology: The analysis is descriptive. Consider applying a standardized histological scoring system for inflammation, fibrosis, and epithelialization.

4. Microbiology: Only Staphylococcus spp. were reported. Please clarify whether further bacterial identification or quantification (e.g., CFU counts) was performed.

5. Outcome measures: Swelling and elevation are crude measures. Please discuss or add biochemical (e.g., hydroxyproline) or biomechanical (e.g., tensile strength) endpoints if available.

6. Discussion: The discussion is repetitive and should be expanded to compare findings with other herbal wound therapies (turmeric, honey, aloe vera, silver dressings). Please highlight translational implications.

Minor Comments

1. Revise the manuscript for conciseness and clarity. Several phrases are repeated unnecessarily.

2. Use scientific names consistently (Mimosa pudica, Cinnamomum verum) alongside local names.

3. Improve figure legends with quantitative data.

4. Ensure reference formatting is consistent with journal style.

Recommendation: Major Revision

6. PLOS authors have the option to publish the peer review history of their article (what does this mean?). If published, this will include your full peer review and any attached files.

Reviewer #1: **Yes:** Rahim Mohammadi

Reviewer #2: **Yes:** Abdulrahman Almalki

---

## [Author Response · Author response to Decision Letter 1]

19 Nov 2025

Dear Editor,

Thank you for considering our paper for revision and resubmission for publication. Please find our detailed answers to your and the reviewer’s comments and suggestions below. We have included most of the suggestions. Please note the highlighted part in the manuscript for correction.

Kind regards

Rukhsana Amin Runa

Comments to the academic editor

Authors response:

We have prepared our revised manuscript by following the instructions of the journal.

2. In your Methods section, please include a comment about the state of the animals following this research. Were they euthanized or housed for use in further research? If any animals were sacrificed by the authors, please include the method of euthanasia and describe any efforts that were undertaken to reduce animal suffering.

Authors response:

We have addressed it in the methods section (see lines 113-116)

The authors appreciate the financial support received from the National Science and Technology (NST), Ministry of Science and Technology (MoST), Government of the People’s Republic of Bangladesh.

Authors response:

We have stated it to the cover letter.

Authors response:

We have given all our raw data and results of statistical analysis that we used for making the graphs and tables, and shared them as a minimal data set in the resubmission process.

Authors response:

We have included the recommended publications of the reviewer in our manuscript.

Comments to the reviewers:

We would like to thank this reviewer for the constructive comments and very helpful suggestions. Specific answers to this reviewer’s suggestions and comments are below:

Reviewer # 1:

The manuscript presents an interesting and relevant study on the comparative efficacy of M. pudica (Lajjabati) and C. verum (Daruchini) extracts on wound healing in rabbits. The topic is important for ethnomedicine and could contribute useful data to the field of herbal wound management. The experimental design is generally sound, and the results are clearly presented with supportive histopathological and bacteriological data. However, some aspects of the manuscript require clarification, additional methodological details, and improvements in presentation to improve the rigor and clarity of the study. This section could be strengthened by citing following literature to better contextualize the study:

-https://doi.org/10.30500/ivsa.2025.519077.1444

-https://doi.org/10.30500/ivsa.2024.428433.1382

Authors response:

Thank you for your constructive and detailed review of the manuscript. We are pleased that you found the study interesting, relevant, and potentially a useful contribution to the field of herbal wound management and ethnomedicine.

To improve its rigor and clarity, we have addressed the points raised and revised the introduction by integrating and citing the recommended literature (See lines 73-81).

Reviewer #1:

Introduction

The introduction provides a good overview of wound healing and traditional herbal treatments. It would benefit from more details on the rationale for choosing these two plants, especially the expected synergistic or differential effects. Some recent references on herbal wound healing could be included to strengthen the background.

Authors response:

We have added more text on the rationale of this study (See lines 87-96)

Reviewer #1:

Results

The morphological, bacteriological, and histopathological findings are clearly described. The graphical figures related to bacteria colony counts could be supplemented with statistical comparisons to highlight significant differences. The histopathology images support the text well, although higher resolution images with scale bars would improve visual interpretation.

Authors response:

We have corrected and added results regarding morphological characteristics, bacterial colony count in the results section (lines 189-190, 199-200, 221-223), and shared our data set with statistical analysis as a supporting file (S1 Table, S2 Table, and S1 Appendix). The resolution of the histopathological images with scale bars has been improved.

Reviewer #1:

Discussion and Conclusion

The discussion appropriately relates findings to previous research. The statement regarding Daruchini causing delayed healing is interesting, but should avoid overgeneralization; it might be due to irritation or tissue reactions, but further mechanistic studies are warranted. Consider emphasizing the practical implications and potential limitations, such as small sample size or animal model limitations. Suggest potential future directions for validation in other models or clinical settings.

The conclusion is consistent with the results and highlights the potential of Lajjabati extract as a wound healing agent. Slightly soften conclusions about Daruchini, as “antagonist” effects require more evidence.

Authors response:

We have addressed all the comments in the discussion and conclusion sections (see lines 323-338, 343-353)

Reviewer #2:

This manuscript explores the efficacy of Mimosa pudica and Cinnamomum verum extracts in wound healing in rabbits. The topic is relevant, and the experimental design (four groups, multiple endpoints, histology, and microbiology) is appropriate. The results suggest that Mimosa pudica has superior wound healing properties, while Cinnamomum verum may delay healing when used alone or in combination. However, the study has methodological weaknesses that need to be addressed before it can be considered for publication.

Authors response:

Thank you so much for your valuable comments. In the methodology, we have addressed several points according to the comments of the editor and reviewers (see lines 113-116, 162, 164, 166, 179-182)

Reviewer#2:

Major Comments

1. Sample size: Only four animals per group were used. Please justify the sample size with a power analysis or acknowledge this as a limitation.

Authors response:

We recognize the concern regarding the sample size. We used four animals per group (in total, 8 wounds per group), which was deemed sufficient for preliminary statistical analysis. However, we concur with the reviewer that this number means the study has reduced statistical sensitivity. We have explicitly acknowledged the limited sample size as a primary limitation in the Discussion section.

Reviewer #2:

Statistical analysis: Provide exact p-values and confidence intervals for all comparisons. Clarify group differences in tables. Healing scores should include inter-rater reliability or a validated scoring system.

Authors response:

Results in Tables 1 and 2 have been corrected, and the dataset with statistical analysis has been provided in the supplementary information (S1 Table, S2 Table, and S1 Appendix). Reference has been inserted into the text for validation of the scoring system (line 162)

Reviewer #2:

Histopathology: The analysis is descriptive. Consider applying a standardized histological scoring system for inflammation, fibrosis, and epithelialization.

Authors response:

We appreciate the reviewer's recommendation to apply a standardized histological scoring method, as it would surely improve the data's quantitative robustness. However, the limited remaining tissue sections, it is not now technically possible to use a pre-established, validated scoring system in a reliable and reproducible manner without sacrificing the analysis's integrity.

Reviewer #2:

Microbiology: Only Staphylococcus spp. were reported. Please clarify whether further bacterial identification or quantification (e.g., CFU counts) was performed.

Authors response:

Thank you for raising this point. The initial report only mentioned the genus Staphylococcus spp.; however, subsequent, more detailed identification confirmed that the exact species was Staphylococcus aureus. The bacterial colony count per 100 microlitre was indeed performed, and these quantitative data are presented in the results section (lines 221-223).

Reviewer #2:

Outcome measures: Swelling and elevation are crude measures. Please discuss or add biochemical (e.g., hydroxyproline) or biomechanical (e.g., tensile strength) endpoints if available.

Authors response:

We completely agree that biochemical (e.g., hydroxyproline) and biomechanical (e.g., tensile strength) endpoints provide superior, more quantitative measures of tissue quality and functional recovery compared to gross measures like swelling and elevation. Unfortunately, since the animal phase of the experiment concluded several months ago, we are unable to retrieve the original tissue samples to perform these retrospective destructive analyses. To address this vital point, we have significantly revised the Discussion section (lines 333-338) to explicitly state this limitation, and we will incorporate your suggestion into our future work.

Reviewer #2:

Discussion: The discussion is repetitive and should be expanded to compare findings with other herbal wound therapies (turmeric, honey, aloe vera, silver dressings). Please highlight translational implications.

Authors response:

We have addressed this and added more text comparing other herbal therapies and translational implications (see lines 308-313, 323-329)

Reviewer #2:

Minor Comments

Revise the manuscript for conciseness and clarity. Several phrases are repeated unnecessarily.

Authors response:

We have gone through the whole manuscript and corrected the errors.

Reviewer #2:

Use scientific names consistently (Mimosa pudica, Cinnamomum verum) alongside local names.

Authors response:

Corrected

Reviewer #2:

Improve figure legends with quantitative data.

Authors response:

Corrected

Reviewer #2:

Ensure reference formatting is consistent with journal style.

Authors response:

Checked and corrected

---

## [Decision Letter · Decision Letter 1]

12 Dec 2025

PONE-D-25-41676R1Efficacy of Lajjabati (Mimosa pudica) and Daruchini (Cinnamomum verum) extracts on wound healing in rabbitsPLOS One

Dear Dr. Runa,

Thank you for submitting your manuscript to PLOS ONE. After careful consideration, we feel that it has merit but does not fully meet PLOS ONE’s publication criteria as it currently stands. Therefore, we invite you to submit a revised version of the manuscript that addresses the points raised during the review process.

We look forward to receiving your revised manuscript.

Kind regards,

Vinod Ayyappan, Ph.D., PostDoc

Academic Editor

PLOS One

Journal Requirements:

Additional Editor Comments:

The authors must address all the comments accordingly

Reviewers' comments:

Reviewer's Responses to Questions

**Comments to the Author**

1. If the authors have adequately addressed your comments raised in a previous round of review and you feel that this manuscript is now acceptable for publication, you may indicate that here to bypass the “Comments to the Author” section, enter your conflict of interest statement in the “Confidential to Editor” section, and submit your "Accept" recommendation.

Reviewer #1: All comments have been addressed

Reviewer #2: (No Response)

2. Is the manuscript technically sound, and do the data support the conclusions?

Reviewer #1: Yes

Reviewer #2: (No Response)

3. Has the statistical analysis been performed appropriately and rigorously? 

Reviewer #1: Yes

Reviewer #2: (No Response)

4. Have the authors made all data underlying the findings in their manuscript fully available?

Reviewer #1: Yes

Reviewer #2: (No Response)

5. Is the manuscript presented in an intelligible fashion and written in standard English?

Reviewer #1: Yes

Reviewer #2: (No Response)

6. Review Comments to the Author

Reviewer #1: (No Response)

Reviewer #2: This manuscript presents interesting preliminary findings; however, several major methodological clarifications and revisions are required. Issues include insufficient detail on randomization and blinding, lack of standardized histological scoring, absence of functional wound-healing measures, inconsistencies in reported results, and overinterpretation of findings beyond the acute rabbit model. With appropriate revision and clarification, the paper could make a meaningful contribution. Therefore, I recommend Major Revision.

7. PLOS authors have the option to publish the peer review history of their article (what does this mean?). If published, this will include your full peer review and any attached files.

Reviewer #1: **Yes:** Rahim Mohammadi

Reviewer #2: **Yes:** Abdulrahman Almalki

---

## [Author Response · Author response to Decision Letter 2]

24 Dec 2025

Dear Editor and Reviewer,

We would like to thank the reviewer for the constructive and thoughtful feedback on our manuscript. We have carefully addressed each concern to improve the scientific rigor and clarity of the study. Please find our detailed answers to the reviewer’s comments and suggestions below. Please note the highlighted part in the manuscript for correction.

Kind regards

Rukhsana Amin Runa

Review Comments to the Author

Reviewer #2:

This manuscript presents interesting preliminary findings; however, several major methodological clarifications and revisions are required. Issues include insufficient detail on randomization and blinding, lack of standardized histological scoring, absence of functional wound-healing measures, inconsistencies in reported results, and overinterpretation of findings beyond the acute rabbit model. With appropriate revision and clarification, the paper could make a meaningful contribution. Therefore, I recommend Major Revision.

Authors response:

Below are our point-by-point responses:

1. Comment: Insufficient detail on randomization and blinding.

Response: We agree that these details are vital for transparency. We have updated the Materials and Methods section (lines 118-120) to explicitly state that rabbits were assigned to groups using a random lottery system. Additionally, we have clarified that the histopathological analysis was performed by investigators blinded to the treatment groups to prevent observational bias.

2. Comment: Lack of standardized histological scoring.

Response: We have added the descriptive observations with a semi-quantitative scoring system (0–3 scale) for inflammation, fibrosis, and glandular changes in the materials and methods section (lines 181-184). This new data has also been added as Table 4 in the results section (lines 238-245).

3. Comment: Absence of functional wound-healing measures.

Response: We acknowledge that biomechanical data (such as tensile strength) would strengthen the study. Since this was a pilot study, we focused on morphological and histo-architectural changes. We have now included this as a specific limitation in the Discussion section and highlighted it as a priority for our follow-up research (lines 352-354; 375-377).

4. Comment: Inconsistencies in reported results.

Response: We have clarified this in the result (lines 199-201; 206-207) and discussion section (lines 311-316)

5. Comment: Overinterpretation of findings beyond the acute rabbit model.

Response: We have toned down the conclusions throughout the manuscript. We have also added a section on limitations regarding the sample size (n=4) and the need for broader clinical trials (lines 365-377).

---

## [Decision Letter · Decision Letter 2]

25 Jan 2026

Efficacy of Lajjabati (Mimosa pudica) and Daruchini (Cinnamomum verum) extracts on wound healing in rabbits

PONE-D-25-41676R2

Dear Dr. Runa,

We’re pleased to inform you that your manuscript has been judged scientifically suitable for publication and will be formally accepted for publication once it meets all outstanding technical requirements.

Kind regards,

Vinod Ayyappan, Ph.D., PostDoc

Academic Editor

PLOS One

Additional Editor Comments (optional):

The manuscript can be accepted in its current form

Reviewers' comments:

Reviewer's Responses to Questions

**Comments to the Author**

1. If the authors have adequately addressed your comments raised in a previous round of review and you feel that this manuscript is now acceptable for publication, you may indicate that here to bypass the “Comments to the Author” section, enter your conflict of interest statement in the “Confidential to Editor” section, and submit your "Accept" recommendation.

Reviewer #2: (No Response)

2. Is the manuscript technically sound, and do the data support the conclusions?

Reviewer #2: (No Response)

3. Has the statistical analysis been performed appropriately and rigorously? 

Reviewer #2: (No Response)

4. Have the authors made all data underlying the findings in their manuscript fully available?

Reviewer #2: (No Response)

5. Is the manuscript presented in an intelligible fashion and written in standard English?

Reviewer #2: (No Response)

6. Review Comments to the Author

Reviewer #2: (No Response)

7. PLOS authors have the option to publish the peer review history of their article (what does this mean?). If published, this will include your full peer review and any attached files.

Reviewer #2: **Yes:** Abdulrahman Almalki

---

## [Editor Report · Acceptance letter]

PONE-D-25-41676R2

PLOS One

Dear Dr. Runa,

I'm pleased to inform you that your manuscript has been deemed suitable for publication in PLOS One. Congratulations! Your manuscript is now being handed over to our production team.

Kind regards,

on behalf of

Dr. Vinod Ayyappan

Academic Editor

PLOS One